# Analysis of the Factorial Structure and Reliability of the Social Determinants of Mental Health Questionnaire for Young Adults (SDMH)

**DOI:** 10.3390/ejihpe15020018

**Published:** 2025-02-02

**Authors:** Monica Roncancio-Moreno, Rita Patricia Ocampo-Cepeda, Walther M. Zúñiga, Arcadio de Jesús Cardona-Isaza

**Affiliations:** Department of Psychosocial Studies, Faculty of Psychology, Universidad del Valle, Campus Universitario Melendez, Cali 760042, Colombia; rita.ocampo@correounivalle.edu.co (R.P.O.-C.); walther.zuniga@correounivalle.edu.co (W.M.Z.); arcadio.cardona@correounivalle.edu.co (A.d.J.C.-I.)

**Keywords:** young adults, mental health, social determinants of health, psychometric analysis

## Abstract

Background: Several studies around the world report an increase in mental health problems among young people. Psychology is called upon to design ways to characterize these problems to make contextualized and effective interventions. The aim of this study was to analyze the factorial structure and reliability of the Social Determinants of Mental Health Questionnaire for Young Adults (SDMH), which was developed based on the Social Determinants of Health Model. Methods: This study included 1232 young Colombians aged 18 to 28 years (M = 20.88; SD = 3.52; 46.8% were women). The questionnaire design was rigorous and involved the participation of experts on the subject, peer review, and a pilot study. Statistical analyses included descriptive, reliability, exploratory, and confirmatory factor analyses. Results: The analyses indicate high reliability, and the confirmatory factor analyses reveal an adequate factorial structure. Conclusions: These findings suggest that the SDMH is a reliable and valid instrument for assessing the social determinants of mental health among young Colombians. Additional studies are needed to consolidate evidence of internal structure validity and provide more information on other sources of evidence regarding test validity.

## 1. Introduction

In recent years, mental health problems among young people have increased, exacerbated by the COVID-19 pandemic ([3]; [10]; [12]; [14]; [24]; [44]). While some young people with adequate personal, family, and material conditions can handle the situation, many others have suffered a serious deterioration in their quality of life ([1]; [8]; [34]). Longitudinal studies conducted with measures before and after the COVID-19 show mixed evidence for symptom constancy, with increases or decreases in conditions such as anxiety and depression ([13]; [15]; [18]; [19]). For example, a study conducted by [16] ([16]) on college students shows that there was an increase in mental health symptoms during the first four months of the pandemic. However, eighteen months later, symptoms associated with depression returned to similar to pre-pandemic values, while anxiety symptoms remained elevated. In addition, the research of [9] ([9]), conducted on college students in 2018–2019 and 2021, identified that participants who reported a higher impact from the pandemic showed, 18 months later, an increase in stress and a decrease in life satisfaction. On the other hand, [18] ([18]), with a sample of adults during the first year of the pandemic, found constant levels of anxiety, perceived stress, and worry but a significant decrease in post-traumatic symptoms by the end of this first year. While longitudinal studies point to the resilience of subjects to recover after the impact of the pandemic, it is also clear that some have remained at alarmingly high levels since then.

Also, most studies report an increase in depression, anxiety, and psychological distress, as well as greater negative affect, poorer emotional well-being, and increased loneliness and other emotional disorders ([21]). This panorama coincides with the reports presented by various international organizations ([43]; [30]) that highlight great needs in mental health; however, the initiatives to mitigate these shortcomings are still insufficient.

In Colombia, the data show a similar trend. The most far-reaching mental health assessment carried out in the country was the National Mental Health Survey (ENSM), which was applied in 2015. In this survey, young people were included in the sample of participants between 18 and 44 years old, and the instruments used corresponded to various scales used to measure the adult population. Adapted questionnaires and other surveys designed by the research team were used to complement the information ([32]).

The ENSM evaluated the sociodemographic aspects, education, mental health status, mental problems and disorders, and access to mental health services of people between 18 and 44 years old. The results revealed high exposure to risk factors in this group ([32]).

Another study on mental health in the Colombian population older than 18 years was PSY-COVID ([33]), which was developed for 30 countries and measured components of sociodemographic information, public health, and mental health. For the study, various adapted instruments were used. In terms of the negative effects on the mental health of young people during the pandemic, a higher prevalence of depression, somatization, loneliness, and low resilience was reported in this population group.

In accordance with the above, in Colombia, there are few data on the mental health of young people; however, recent evidence shows a growing deterioration of mental health in this group ([33]). Faced with this reality, it is important to have precise and contextually adequate tools to assess the mental health of young people. For this reason, the Social Determinants of Mental Health Questionnaire for Young Adults (SDMH) was designed as an instrument that seeks to capture the complexities of mental health in this population and integrate various key dimensions in the present.

Regarding the instruments used to evaluate mental health in young people, the systematic review by [23] ([23]) revealed that the commonly used instruments have not been designed specifically for young people (see the review of instruments by [23]). Thus, it is important to design measures that are sensitive to reality and the ongoing changes in this population in this crucial period.

This article presents a questionnaire that was developed to inquire about the mental health situation of young people from a salutogenic perspective. The main objective is to analyze the factorial structure and reliability of the SDMH, developed from the Model of Social Determinants in Health ([42]).

The instrument was designed based on a definition of mental health supported by information from the WHO, the public policy on mental health and ([25]) in Colombia, which is understood as the resources and strategies (personal, collective, emotional, intellectual, and behavioral) that are used on a daily basis to cope with the challenges and demands of life and the associated emotional experience of well-being or discomfort. In this definition, the adequate development and enjoyment of social activities are of great personal importance, as are meaningful interpersonal relationships, work, study, and participation in recreational and cultural activities. The degree of satisfaction with the development and outcome of these personal and social activities is considered to have an impact on mental health. Mental health is thus dynamic and complex, referring to specific moments and circumstances delimited in time.

The instrument was theoretically based on the WHO Model of Social Determinants in Health (2009), which refers to the “structural determinants and living conditions that are the cause of much of the health inequities between countries and within countries” (p. 1). This model has been adopted by the WHO and used in various investigations; thus, it constitutes a solid perspective for the construction of the SDMH ([17]; [35]; [40]). In particular, the model specifies structural and intermediate determinants. Structural determinants are those that “generate or reinforce stratification in society and that define individual socioeconomic position” ([31]), in which gender, race, and ethnicity are classified. Education, occupation, and income depend on socioeconomic position. In contrast, intermediate determinants are defined as “the intermediate factors between structural determinants and the unequal distribution of health and well-being in the population” ([31]). The intermediate determinants correspond to the material conditions of life, the psychosocial context, social cohesion, lifestyles, biological factors, people’s behaviors, and the health system.

In accordance with all of the above, the instrument investigates the structural and intermediate social determinants that, among other factors, determine a young person’s living situation, the challenging/stressful situations that he faces, the way in which he copes, and the subjective experience of well-being and discomfort in different periods of time. Risk factors and protective factors are also explored. Figure 1 presents the theoretical elements of the questionnaire and their relationships.

A relevant contribution of this study is the development of an instrument that is based on the Model of Social Determinants and, in turn, considers cultural and contextual aspects to assess mental health from a positive and salutogenic perspective, as indicated by public policy on the subject of mental health in Colombia ([25]; [27]). This instrument can be useful both for individual assessment of young people and for mental health research.

## 2. Method

### 2.1. Participant Characteristics

The final sample of the study consisted of 1.232 young university students residing in the city of Cali, Colombia, aged between 18 and 28 years (M = 20.48 years, SD = 2.26), of whom 46.8% (*n* = 576) were women. A total of 72.6% identified themselves as mestizos, 20.8% as Afro-descendants, 6.4% as indigenous, and 0.2% as Roma. In the area where the study was carried out, there is wide ethnic and cultural diversity, and various groups coexist, including the Afro-descendant community, which is widely represented, and a significant part of the almost one hundred indigenous groups of the country, each of which has its own culture, language, and ancestral laws, and whose peoples reside in special territories known as resguardos. A total of 93.8% of the participants were studying between the first and fifth university semesters in various disciplines, mainly engineering, health, and political science. With respect to sexual orientation, 86.5% identified themselves as heterosexual, 0.8% as homosexual, and 9.9% as bisexual, and the rest indicated other orientations or preferred not to use labels. Other sociodemographic aspects are detailed in Table 1.

### 2.2. Inclusion and Exclusion Criteria

The inclusion criteria were men and woman aged between 18 and 28 years. The exclusion criteria included people outside of this age range.

### 2.3. Sampling Procedures

This study comprised a convenience sample of young people from different social backgrounds with diverse socioeconomic conditions that represented the cultural multiplicity of the country, belonging to a public university. An open call was made to all young adults in the university community and outside who wished to participate voluntarily in the study through various mechanisms, such as social networks, email, and WhatsApp. Any young adult within the age range could participate.

### 2.4. Instrument

The SDMH is a 72-question instrument designed to assess determinants of mental health, and is based on the model of health determinants accepted and promoted by the WHO. The determinants of mental health are a complex amalgam of individual, social, and structural factors that interact to shape people’s social and psychological well-being.

The SDMH is of a mixed nature, and has qualitative components in which people can elaborate their answers in detail; for example, it includes questions such as “Who do you entrust your concerns or problems to?”, “In your words, how do you feel? What is your mental health today?”, and “Mention the three activities that you enjoy the most, that fill you with vitality, joy and/or satisfaction”. The forms of response can be free, dichotomous, multiple choice or Likert-type ordinal. Higher scores for risk factors indicate greater negative impacts, and high scores for protective factors indicate greater favorability.

Structural factors include social and economic conditions; access to health systems; and ethnic, school, and labor issues. The intermediate *determinants* in the questionnaire are evaluated qualitatively and quantitatively, and are composed of the factors of well-being in the environment, protective factors, risk factors, well-being, and discomfort in the last week and in the last six months.

The questionnaire includes an evaluation of risk factors that correspond to intermediate determinants related to physical and psychological health problems. These factors disadvantage mental health, and include emotional concern and discomfort, suicidal behavior, drug use, addictions, and related problems.

Protective factors for well-being and mental health, which are also intermediate determinants, include well-being in different settings, the expectation of social support, healthy lifestyle habits, and psychosocial life skills.

In the outcome factors section, emotional and physical well-being and discomfort are evaluated for the past week and the past six months. This section assesses the presence of symptoms associated with mental health problems, and high scores indicate the need for specialized care and assessment. The descriptive and reliability data for each of the factors are presented in Table 1 and Table 2.

### 2.5. Measurement of Constructs and Procedure

The construction of the instrument was carried out with reference to the ten steps proposed by [29] ([29]). The questionnaire is based on the WHO Model of Social Determinants of Health, and is aligned with the Colombian regulatory framework and public policy documents on mental health ([25]; [27]).

The dimensions and their variables were defined, specifying both structural and intermediate determinants, and items were designed to align with these constructs. The instrument evaluates the social determinants of mental health, organized into two main domains, according to the WHO model (2009): structural determinants and intermediate determinants. Within these domains, specific dimensions were identified, grouping subdimensions and items designed to capture the most relevant aspects of each category. Appendix A provides examples of the questionnaire items and their scores. It also provides the questionnaire’s structure, indicating the corresponding determinant for each question and the component or factor.

An interdisciplinary team wrote and edited the items, incorporating adjustments derived from expert reviews. Additionally, pilot studies were conducted, and the test was administered. The psychometric properties were then analyzed through reliability studies, content validity, exploratory factor analysis (EFA), and confirmatory factor analysis (CFA), ensuring the instrument met established standards.

The structural determinants domain includes factors that generate social stratification and define individuals’ social, economic, vital, and cultural positions. The dimensions evaluated within this domain involve sociodemographic conditions, access to health services, and vital transitions. On the other hand, the intermediate determinants domain addresses material, behavioral, and psychosocial factors that directly affect mental health. This domain includes well-being factors, protective and risk factors, and outcome variables related to well-being and distress, which were evaluated over the last week and the last six months.

The questionnaire considers differential approaches and vital transitions during life. Also, it emphasizes protective and risk factors, and the subjective experience of well-being (discomfort), which will be explained below:

Protective Factors. These factors include the psychosocial skills for life proposed by the [41] ([41]), the healthy habits and activities that young people enjoy, the way they cope with stressful situations, and their dispositions and expectations regarding seeking help from others. Additionally, the questionnaire recognizes that daily life takes place in different environments, such as the neighborhood, the workplace, and educational institutions, and the way people feel in these places and the kinds of relationships they establish play very important roles in mental health, well-being, and quality of life. The questionnaire investigates the quality of relationships, presence of conflicts, forms of resolution, and general perception of satisfaction in different environments.

Risk Factors. In terms of risk factors, those investigated are the situations (and areas) of greatest concern to young people, the presence of physical or mental illness, the presence of emotional injuries (trauma) and their duration, situations of violence, problematic consumption, problematic internet use, and suicidal behavior.

Subjective Experience of Well-being (Discomfort). A basic screening of mostly negative experiences (symptoms) associated with mental health and well-being is incorporated. This inquiry is made for two specific time scenarios (last six months and last week).

Application specifications. The questionnaire was designed to be applied individually and collectively through computer media or on paper.

About the items. The first version of the instrument consisted of 90 questions with various response modalities (closed dichotomous responses, closed responses with 3 or 5 options, and open responses). In each section, descriptions were included to clarify the meaning and purpose of the questions, in order to reduce misunderstandings and encourage the participant to answer as honestly as possible.

Construction of the items. To exhaustively address each of the aspects identified for inclusion, the number of questions is not necessarily the same in each section. The guidelines and recommendations of the National Department of Statistics ([11]) were followed regarding sociodemographic data, including to how to inquire about sexual orientation, ethnicity, and gender identity.

The items were written and reviewed by the team of researchers, and recommendations and observations were incorporated until a version accepted by the entire team was obtained. The items were written in clear and inclusive language.

To evaluate the clarity of the items, in terms of writing, vocabulary, and other aspects that could generate confusion, a first pilot was carried out with 96 psychology and medicine students who read the questionnaire in its entirety and indicated errors in the typing, writing, or clarity of the questions. All errors were corrected, and the clarity of the wording was improved in each case indicated.

Finally, five local experts were consulted to ensure coherence with the social, economic, and cultural context: professionals in psychology and medicine, and specialists in mental health issues (three) and/or instrument design (two). The experience of each of the experts is described below.

Expert 1: Psychiatrist, doctor in psychology, university professor. She participated in the design of the Colombian National Mental Health Survey.

Expert 2: Psychologist, master’s degree in psychology, university professor in clinical psychology, expert in mental health and health psychology.

Expert 3: Psychologist, master’s degree in medical psychology, doctoral candidate, clinical psychologist and university lecturer.

Expert 4: Psychologist, master’s degree in clinical psychology, expert in the design of psychometric instruments.

Expert 5: Psychologist, master’s degree in Psychodiagnosis and Psychological evaluation, university professor.

The experts were asked to evaluate the clarity (in the grammatical construction of the questions, related to aspects such as writing, punctuation, spelling, and the use of adequate and appropriate terms, given the age of the target population), relevance (degree of correspondence or coherence between what the item investigates and the characteristic/category of interest), and type of response (whether the response options adequately reflect the intended subject of investigation) on a dichotomous scale (yes/no). The content validity index (CVI) was calculated for each item, considering the agreement among the experts. A proportions analysis was included to evaluate the items’ clarity and relevance, with an acceptance criterion of at least 80% consensus among the experts. Similarly, they were asked to make observations, explain the reasons supporting their assessment, and suggest a reformulation of the question or the characteristics that a new version of the item should have. Finally, they could make a general comment on the questionnaire. For greater rigor and completeness, all the observations were reviewed (not only those in which there was agreement between the experts), the arguments presented were examined, and a response was given to each of them, indicating whether the item was changed, deleted, or incorporated, or giving explicit support for why the item should be kept in its current version.

Pilot study. A qualitative pilot study was carried out to identify problems with the clarity of the items. This made it possible to identify the average response time for the entire instrument, and the possible drawbacks of its completion via mobile devices.

Ethical aspects. The research followed ethical considerations as provided in country regulations. It was approved by the Ethic Committee from Faculty of Psychology (CEIFP) of the Universidad del Valle, with the consecutive number 23 from 9 March 2023. All the participants signed an informed consent form describing the extent of their participation.

### 2.6. Data Collection

Participants were brought into classrooms to answer the online questionnaire via computers, tablets, or mobile devices between August and November 2023. No time limit was given to participants. A certified psychologist accompanied the filling out of the questionnaire and was available to answer any questions or technical difficulties with the administration of the instrument.

### 2.7. Analytic Strategy

First, a descriptive analysis of the study variables, including the mean and standard deviation, was conducted. The reliability of the quantitative intermediate determinant factors was assessed using Cronbach’s alpha and McDonald’s omega (Table 2). Similarly, correlation analyses were carried out between the risk and protection factors via the SPSS v.26 program ([20]). Two multiple regressions were also performed, using, as dependent variables, two outcome factors: emotional well-being and emotional symptoms in the last six months. The independent variables were well-being factors in the environments, risk factors, and protective factors. The confirmatory factor analysis (CFA) for each of the factors was performed via the AMOS Version 24 program, and analyses were performed with the maximum likelihood estimate.

In this study, a sample size calculation was performed to ensure the robustness of the CFA of the SDMH using the sample size calculator for structural equation models (SEMs) from Analytics Calculators. The model considered 21 latent and 115 observed variables, with a strict significance level (*p* = 0.01), an expected effect size of 0.3, and a desired statistical power of 0.99. The calculation indicated a minimum sample size of 459 participants, suggesting that the obtained sample exceeded the required minimum ([28]).

The general adjustment indices used in this study include the comparative fit index (CFI) ([4]), the Tucker–Lewis index (TLI) ([39]), and the root mean square error of approximation (RMSEA) ([37]). The commonly used guidelines that indicate an acceptable fit of the data to the model for these indices are the following: (a) a CFI and TLI greater than 0.90 ([0.90 = good fit]); (b) an RMSEA of less than 0.08 (<0.05 = good fit, <0.08 = fair fit, <0.10 = poor fit) ([5]). Previous authors also recommend that the factor loadings of the items be of reasonable magnitude, with absolute values greater than 0.30 ([38]). The study analyzed the SDMH’s configurational, metric, scalar, structural, and residual invariance by gender (male = 0, *n* = 656; female = 1, *n* = 576). The configurational invariance indicates whether the factor structures are identical, i.e., the same number of factors and items have the same free and fixed loading patterns. We also assessed the metric invariance, which indicates the degree of equality of the factor loadings on the items, and the scalar invariance, which shows the degree of equality of the factor loadings and the residuals. In addition, the structural invariance was evaluated, which indicates the degree of equality of the variances and covariances of the group factors. Finally, the residual invariance shows whether the variances of the errors are equal.

The invariance was evaluated based on the [7] ([7]) criterion, which indicates that if the difference between the comparative fit indices (CFI) of the compared models is less than 0.01, invariance can be considered to exist. The observed results indicate configurational, metric, scalar, structural, and residual invariance by gender.

## 3. Results

The SDMH investigates various aspects that are included in the Model of Social Determinants of Health of the WHO. The structural determinants include work and affiliation with the health system; these questions are answered as yes or no. For intermediate determinants, some questions have yes or no answers, and others allow more details to be given if the answer is yes. This is the case, for example, in diagnoses of physical and mental health, exposure to stressful events, and exposure to various forms of violence. These data are important because of their usefulness for assessing risk to mental health and well-being. Table 1 shows the answers to some of these questions distributed by gender. Correlation analyses were also carried out between the study factors, and the associations presented the expected directions, with positive associations between the factors of each domain and negative associations between protective and risk factors (see Appendix A). We also performed two multiple regression analyses, using two outcome factors as dependent variables: emotional well-being and emotional symptoms in the last six months. The independent variables included factors related to well-being in different settings, risk factors, and protective factors. The main factors that showed effects on emotional distress were fewer healthy lifestyle habits, higher levels of worry and emotional distress, increased suicidal behavior, drug use, and problems associated with problematic internet use. On the other hand, factors that showed positive effects on emotional well-being in the past six months were well-being at home and psychosocial life skills. The detailed results of these regressions can be found in Appendix A.

### 3.1. Reliability

The analyses show that the factors present adequate reliability, except for school well-being (α = 0.38) and problematic use of the internet (α = 0.45), which have lower values than expected. The section of outcome factors, which describes emotional, physical, and well-being symptoms in the last week and the last six months, presents the indicators with the highest reliability (α = 0.78 to 0.94). The reliability values can be found in Table 2.

### 3.2. Evidence of the Factorial Structure of the Questionnaire

The factorial structure of the SDMH was analyzed by means of CFA. Previously, each of the factors and dimensions were analyzed via exploratory factor analysis (EFA) to verify the distribution of the items and confirm that the factor loadings and the explained variance were adequate. Once this previous step was verified, we proceeded directly with the CFA. The first domain to be analyzed was well-being in various settings. For this set of factors, the EFA showed that the domain had an explained variance of 64.13%, and was made up of six factors that address the perception of well-being in an environment: the questions address relationships, conflicts, and their solutions in the family, neighborhood, school, religious association, and social and friendship environments (Kaiser–Meyer–Olkin = 0.797, *χ*^2^ = 15152.40, *df* = 276, *p* > 0.0001). Overall, the 24 items showed high reliability (α = 0.84). Although the goodness-of-fit indices did not reach the optimal cutoff values (Table 3), the high factor loadings and the associations among the six factors indicated the existence of the construct. The lack of optimal values may be due to the high covariance between the items, related to the correlation among conflicts and their resolution between the factors of well-being in the home and in religious organizations. The results of the CFA are summarized in Table 3 and detailed in Appendix A.

The EFA of protective factors showed an explained variance of 56.32% and yielded a grouping of three factors: expectations of social support, healthy lifestyle, and psychosocial skills for life (Kaiser–Meyer–Olkin = 0.874, *χ*^2^ = 10,389.47, *gl* = 325, *p* > 0.001). Overall, the 26 items showed high reliability (α = 0.81). With respect to the factor of expectations of social support, high covariance was observed between the expectations of support from friends and coworkers. Similarly, high covariance was also found in the psychosocial skills factor, between the expression of thoughts and ideas and the expression of feelings and emotions; in these cases, the indices were modified to improve the fit of the model. The results of the CFA are summarized in Table 3 and detailed in Appendix A.

The domain of *risk factors* was also analyzed through EFA, which showed an explained variance of 55.25 and yielded a grouping of six factors (Kaiser–Meyer–Olkin = 0.880, *χ*^2^ = 16,848.79, *gl* = 496, *p* > 0.001). Overall, the 32 items showed high reliability (α = 0.81). High covariance was observed between the items regarding legal problems and the factors of problems due to drug use and problematic use of the internet. The same pattern was observed for the items that referred to work and partner problems. This seems to indicate that both problems with drug use and problematic use of the internet have shared variance, and explain the labor-related, relational, and legal problems derived from these behaviors. A modification of indices was carried out to improve the goodness-of-fit indicators. The results of the CFA are summarized in Table 3 and detailed in Appendix A.

The EFA of the *outcome factors of the last week* revealed that, together, three factors explained 61.01% of the variance (Kaiser–Meyer–Olkin = 0.907, *χ*^2^ = 10326.76, *gl* = 136, *p* > 0.001). The 17 items of the factors showed high reliability (α = 0.86). The EFA of the outcome factors of the last six months revealed that, together, three factors explained 69.936 of the variance (Kaiser–Meyer–Olkin = 0.930, *χ*^2^ = 14,860.01, *gl* = 136, *p* > 0.001). The 17 items of the factors showed high reliability (α = 0.92).

For the outcome factors, the goodness-of-fit indices of the CFAs for physical and emotional well-being and discomfort were adequate, although we observed high covariance in the anxiety and worry items, and a lack of enjoyment and loss of interest in the emotional symptoms factor. This covariance pattern was observed in both CFAs. The results of the CFAs are summarized in Table 3 and detailed in Appendix A.

To evaluate the factorial structure of the questionnaire in the intermediate determinants component, a confirmatory factor analysis (CFA) was conducted, including the 21 factors and all items (*n* = 115). The results indicated that the overall instrument showed moderate fit indices. High factor loadings were observed for the items (Appendix A). Although optimal indicators were not achieved, the findings supported the existence of a factorial structure when including all of the proposed factors (*χ*^2^ = 19,642.11, *df* = 6211, *p* > 0.001, CFI = 0.829, TLI = 0.820, RMSEA = 0.042 [90% CI: 0.041–0.043]).

### 3.3. Invariance Analysis

For the invariance analysis, we selected the full model of the Social Determinants of Mental Health Questionnaire for Young Adults (SDMH), which consists of 115 items and 21 factors. The invariance was evaluated based on the [7] ([7]) criterion, which indicates that if the difference between the comparative fit indices (CFIs) of the compared models is less than 0.01, invariance can be considered to exist. According to the stated criterion, if the global fit indices, such as CFI and RMSEA, are adequate, and the differences in ΔCFI (≤0.01) and RMSEA (<0.08) are small, measurement invariance can be considered to hold, despite statistically significant differences in the chi-square test. The observed results indicate configurational, metric, scalar, structural, and residual invariance by gender (Table 4). The results of the gender invariance analysis can be found in Appendix A.

## 4. Discussion

The objective of this study was to analyze the factorial structure and internal validity of the Social Determinants of Mental Health Questionnaire for Young Adults (SDMH), which was developed from the Model of Social Determinants in Health ([42]) and from a salutogenic perspective. It goes beyond the medical model by considering that there are factors that help people to maintain and improve their well-being in all dimensions.

The results indicate that the SDMH is reliable and has an adequate factorial structure for evaluating the social determinants of mental health in this population. In Colombia, the need for specific instruments to assess and address these problems is crucial, which is why the SDMH could help in research and clinical practice in the Colombian context. The questionnaire addresses a wide range of social determinants that have been identified as affecting mental health, including structural factors, such as access to health systems and economic conditions, and intermediate factors, such as well-being in various settings and psychosocial skills ([6]). This breadth of measurement allows a comprehensive assessment of the mental health of young people, capturing both risk factors and protective factors.

The questionnaire was formulated via a process of reflection and analysis, considering different structured steps ([29]), and was intended to capture structural and intermediate determinants, including ethnic diversity, violence, and social conflicts.

The information addressed in a descriptive way regards previous and current experiences and perceptions, and can infer how certain issues affect people’s well-being, relationships, and mental health ([6]). These items provide relevant information for the prevention and promotion of mental health, and can be used to guide public policies and develop interventions throughout the life course ([22]).

The factorial structure of the SDMH was analyzed through EFA and CFA of the components that were evaluated as factors—specifically, well-being in different environments and risk and protective factors—and the result components focused on emotional and physical symptoms and well-being. The reliability indices (Cronbach’s alpha and McDonald’s omega) were adequate, except for well-being at school and problematic internet use.

Some covariance problems were detected in the CFAs, which suggests the need to review the items and compromised factors for possible future improvements. Procedures such as the modification or elimination of items may help to improve the reliability and psychometric properties of the instrument ([36]). First, it is necessary to investigate whether covariance is due to problems in the items, errors in interpretation, or other reasons; in any case, the items that were included show adequate factor loadings.

The outcome factors are intended to assess emotional, physical, and mental health symptoms in the last week and in the last six months. It is through these factors that the impacts on mental health can be weighed; therefore, they are the main factors of the questionnaire, and can be used to assess whether young people should visit health services for specific evaluations. Both the reliability and the goodness-of-fit indices in the CFAs of these factors were satisfactory.

For the factors included in the CFA, correlation analyses were performed, which revealed that the factors were associated within each domain and with each other in the expected way. This finding indicates that the factors explain the existence of the social determinant construct of mental health, and that the relationships between the factors are consistent with the theoretical hypotheses ([42]).

The main contribution of the SDMH is that it addresses a wide range of social determinants, including structural factors; intermediate factors, such as well-being in various environments; protective and risk factors; and affect throughout the life course of young people.

Despite the positive findings and contributions, there are limitations that must be considered. For example, the sample was composed exclusively of young university students, and additional applications are needed to support the generalization of the results to other contexts or demographic groups. Furthermore, the low reliability of some factors suggests the need to review and possibly reformulate certain items to improve their internal consistency. Another limitation is that the validity of the questionnaire has not been demonstrated via other tests established by the American Educational Research Association (AERA), the American Psychological Association (APA), or the National Council on Measurement in Education (NCME) ([2]). The relationships with other variables and the consequences or impacts of various factors, among other aspects, need to be tested.

The questionnaire shows promising results, but it is necessary to carry out subsequent psychometric studies to provide further evidence of its validity. Similarly, it is proposed to unify the response scale across factors, as this may be a cause of convergence and covariance problems between some items. We consider that the SDMH presented constitutes a good basis for the evaluation of mental health in young people, but it may be improved in terms of the form of measurement, and in some components, for example, in the evaluation of environments. Likewise, future studies should focus on providing more evidence for the validity of the instrument and its application in various populations and contexts. It would also be beneficial to explore its applicability in various contexts and in other regions of the country.

The results of this study have important implications for clinical practice and mental health research. The SDMH can be used by health professionals to characterize, in context, the mental health situation of individuals and groups in a way that includes the social determinants that influence mental health; this can help clinicians to gain a deeper understanding of the complexities and specific needs of the groups they work with, which is crucial for designing effective and personalized interventions. By allowing researchers to explore the relationships between social determinants and mental health, since the questionnaire covers a wide range of social determinants, the results can facilitate a multidisciplinary approach to addressing mental health. This involves collaboration between psychologists, social workers, educators, and public health professionals. This approach can contribute to improving the quality of life of young people and reduce the stigma associated with mental health conditions, promoting greater acceptance and understanding within the community.

In conclusion, the SDMH is a helpful tool for the comprehensive evaluation of the mental health of young people, capturing a wide range of social determinants, risks, and protective factors. This study provides a solid basis for assessing the factorial structure and reliability of the SDMH, developed from the model of social determinants in health ([42]) from a salutogenic perspective. However, it is necessary to improve some aspects, carry out additional analyses, and provide more evidence to support the internal consistency and generalization of the results.

## Figures and Tables

**Figure 1 ejihpe-15-00018-f001:**
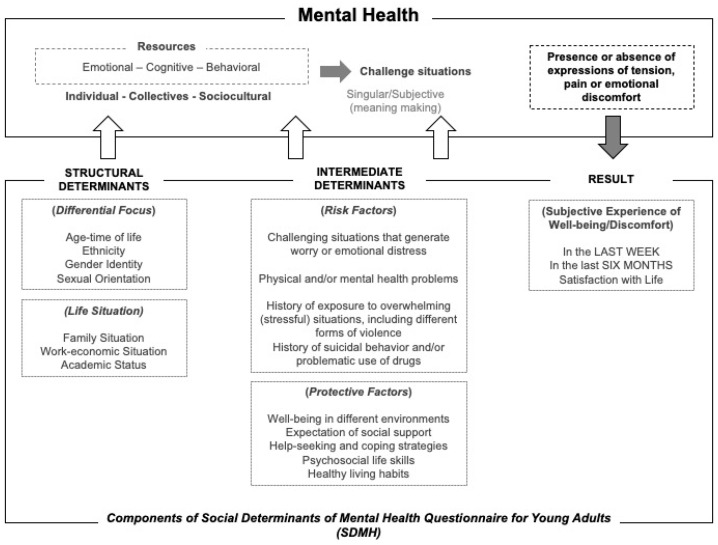
Theoretical components of the Social Determinants of Mental Health Questionnaire for Young Adults (SDMH).

**Table 1 ejihpe-15-00018-t001:** Descriptive statistics of the structural and intermediate determinants not included in the CFA (*n* = 1232).

Variable	Items	Women		Men	
Sociodemographic data					
		*n*	%	*n*	%
Sex		576	46.8	656	53.2
Ethnicity	Mestizo	396	68.8	499	76.1
	Afro-descendant	146	25.3	110	16.8
	Indigenous	33	5.7	46	7.0
	Roma	1	0.2	1	0.2
Structural determinants associated with the vital transition					
Family nucleus	Family of origin	549	95.3	627	95.6
	Other relatives	20	3.5	20	3.0
	Other people, not relatives	2	0.3	4	0.6
	Couple and children	1	0.2	2	0.3
	Single couple	1	0.2	1	0.2
	Alone	3	0.5	2	0.3
Marital status	Single	532	92.4	621	94.7
	Married	9	1.6	3	0.5
	Free union	34	5.9	30	4.6
	Divorced	1	0.2	1	0.2
	Widower			1	0.2
Currently in a relationship	Yes	181	31.4	189	28.8
Study	Yes	576	100	656	100
Satisfaction with studies	Do not know	22	3.8	23	3.5
	Very low	12	2.1	8	1.2
	Low	36	6.3	47	7.2
	Medium	153	26.6	173	26.4
	High	267	46.4	296	45.1
	Very high	86	14.9	109	16.6
Works	Yes	174	30.2	209	31.9
Dependent persons	Yes	44	7.6	55	8.4
Affiliated with health services	Yes	558	96.9	621	94.7
Intermediate determinants					
Chronic health problem	Yes	67	11.6	77	11.7
Treatment effectiveness: health problems	Yes	48	8.3	53	8.1
Functional diversity	Yes	101	17.5	132	20.1
Treatment or rehabilitation	Yes	50	8.7	81	12.3
Diagnosis in mental health	Yes	115	20.0	137	20.9
In treatment: mental health	Yes	67	11.6	69	10.5
Exposure to stressful situations	Yes	473	82.1	530	80.8
	Many years ago	242	42.0	279	42.5
	More than six months ago	122	21.2	125	19.1
	In the last six months	61	10.6	87	13.3
	Currently	46	8.0	36	5.5
Victim of violence	Yes	309	53.6	347	52.9
	Physical	76	13.2	88	13.4
	Sexual	47	8.2	51	7.8
	Psychological	112	19.4	122	18.6
	In social networks	10	1.7	8	1.2
	Bullying	43	7.5	59	9.0
	Forced displacement	21	3.6	19	2.9

Note: These items did not enter the CFA because they are qualitative, categorical, or nominal; they are analyzed descriptively. Additionally, there are questions in the questionnaire that are answered in an open-ended manner.

**Table 2 ejihpe-15-00018-t002:** Descriptive statistics and reliability data for the intermediate determinants included in the CFA (*n* = 1232).

	Factor	Items	Scale Score	Minimum and Maximum Scores	Women (*n* = 576)		Men (*n* = 656)		α	ω
					*M*	*Of*	*M*	*Of*		
Well-being in the environment	Well-being at home	4	0–6	0–24	17.44	3.70	17.54	3.77	0.71	0.71
	Well-being in the neighborhood	4	0–6	0–24	13.72	4.89	13.80	4.93	0.59	0.63
	Well-being in religious organizations	4	0–6	0–24	7.02	7.61	6.26	7.65	0.87	0.88
	Social welfare, culture, and recreation	4	0–6	0–24	8.78	8.64	9.66	8.81	0.90	0.92
	School wellness	4	0–6	0–24	16.74	3.67	17.22	3.50	0.38	0.39
	Well-being at work	4	0–6	0–24	16.50	3.97	16.44	4.10	0.92	0.93
Protective factors	Expectation of social support	8	0–6	0–48	25.79	9.08	26.36	9.16	0.67	0.69
	Healthy lifestyle habits	8	1–6	1–48	25.15	5.88	25.64	6.11	0.71	0.69
	Psychosocial skills for life	10	1–5	10–50	36.19	7.91	36.34	7.88	0.90	0.90
Risk factors	Worry and emotional distress	5	0–4	0–20	10.82	3.75	10.54	3.545	0.66	0.67
	Suicidal behavior	3	0–1	0–3	1.04	1.03	0.97	1.05	0.63	0.63
	Drug use	5	0–4	0–20	2.70	2.68	2.78	2.61	0.63	0.62
	Conflicts due to drug use	7	0–2	0–14	4.37	3.52	4.73	3.56	0.93	0.93
	Problematic internet use	4	0–4	0–16	6.27	2.62	6.73	2.77	0.45	0.55
	Problems due to problematic internet use	7	0–2	0–14	6.31	2.805	6.20	2.94	0.85	0.85
Well-being and discomfort in the last week and the last six months	Emotional symptoms in the last week	8	1–5	8–40	24.08	8.19	23.72	7.86	0.91	0.91
	Physical symptoms in the last week	5	1–5	5–25	14.03	5.31	13.60	4.91	0.82	0.82
	Well-being in the last week	4	1–4	4–20	12.49	3.52	12.93	3.54	0.78	0.79
	Emotional symptoms in the last six months	8	1–5	8–40	23.43	8.17	23.86	8.49	0.94	0.93
	Physical symptoms in the last six months	5	1–5	5–25	13.72	5.5	13.37	5.30	0.88	0.88
	Well-being in the last six months	4	1–4	4–20	12.65	3.81	12.85	3.86	0.85	0.85

**Table 3 ejihpe-15-00018-t003:** The results of the CFA of the intermediate determinants of the mental health questionnaire for young people (SDMH).

Domain	*χ* ^2^	*gl*	CFI	TLI	RMSEA (90% CI)
Well-being in various settings	2127.48	234	0.874	0.851	0.081 (0.078, 0.084)
Protection factors	1173.18	290	0.902	0.913	0.050 (0.047, 0.053)
Risk factors	1721.05	415	0.908	0.918	0.051 (0.048, 0.053)
Well-being and discomfort in the last week	867.78	114	0.912	0.926	0.073 (0.069, 0.078)
Well-being and discomfort in the last six months	997.89	114	0.929	0.940	0.079 (0.075, 0.084)
Total scale (115 items)	19,642.11	6211	0.829	0.820	0.042 (0.041, 0.043)

Note: *n* = 1232 *χ*^2^ = chi-square; *gl* = degrees of freedom; CFI = comparative fit index; TLI = Tucker–Lewis index; RMSEA = root mean square error of approximation. Items of CFA: 27, 28, 29, 30, 36, 37, 38, 62, 63, 64, 65, 66, 71, and 72.

**Table 4 ejihpe-15-00018-t004:** Confirmatory factor analysis (CFA) of the total model and invariance by gender.

	*χ* ^2^	*gl*	*p*	IFI	TLI	CFI	RMSEA (90% CI)	∆*gl*	∆CFI
First-order model, 115 items, 21 factors (*n* = 1232)	19,642.116	6211	<0.001	0.830	0.820	0.829	0.042 (0.041, 0.043)		
Females: First-order model, 115 items, 21 factors (*n* = 576)	13,092.82	6211	<0.001	0.823	0.811	0.821	0.044 (0.043, 0.045)		
Males: First-order model, 115 items, 21 factors (*n* = 656)	13,978.52	6211	<0.001	0.815	0.803	0.814	0.044 (0.043, 0.045)		
Invariance by gender									
Configural invariance	27,071.44	12,422	<0.001	0.819	0.807	0.817	0.031 (0.030, 0.031)		
Metric invariance	27,185.07	12,516	<0.001	0.818	0.808	0.817	0.031 (0.030, 0.031)	113.63	0.001
Scalar invariance	27,331.15	12,631	<0.001	0.818	0.810	0.817	0.031 (0.030, 0.031)	146.08	0.001
Structural invariance	27,653.76	12,863	<0.001	0.816	0.812	0.815	0.031 (0.030, 0.031)	322.61	−0.002
Residual invariance	27,939.72	12,996	<0.001	0.814	0.812	0.814	0.031 (0.030, 0.031)	285.96	−0.001

Note: *χ*^2^: chi-square; *gl* = degrees of freedom; CFI: comparative fit index; TLI: Tucker–Lewis index; RMSEA: root mean square error of approximation; ∆*gl* = change in degrees of freedom; ∆CFI = change in CFI. No changes were observed in ∆RMSEA.

## Data Availability

The research data of the article can be made available upon request to the authors.

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
