# Peer review of "Analysis of the Factorial Structure and Reliability of the Social Determinants of Mental Health Questionnaire for Young Adults (SDMH)"

_ejihpe, 2025, doi:10.3390/ejihpe15020018_

Round 1
Reviewer 1 Report
Comments and Suggestions for Authors
In lines 83 and 84, the authors assert that the model supporting the questionnaire has been utilized in several studies; however, no references are provided to substantiate this claim.
According to the constructions of the items, several procedures were conducted but could be important to name every procedure. Besides, due to the use of yes/no answers to valuate clarity or relevance of the items, a quantitative validity procedure could be reported.
Based on the ethical aspects, it is important to name the institution that provided the approval.
In table 1 it is not clear why some items like Works, Dependent persons or Study had only “yes” answers. Could be necessary to describe how was used the answers to create the variable values to calculate x2, CFI, TLI, RMSEA (addition? average?)
Although the questionnaire has shown good psychometrics fits, it is important to consider broad up the determinants analysis (regression in supplementary file) based on the name of the questionnaire.
In supplementary file, table AIC columns BIC and CAIC has no data information. Finally, in the Nested Model Comparisons “p” use capital letters and could be confuse with another statistic.
Author Response
Dear reviewer, thank you for your accurate comments, please see the attachment.

Reviewer 2 Report
Comments and Suggestions for Authors
The authors conducted a commendable research study analyzing the factorial structure and reliability of the Social Determinants of Mental Health Questionnaire for Young Adults (SDMH). This work is an important contribution to the growing body of literature on the intersection of social determinants and mental health, particularly among young adults. While the study is limited to young adults in Columbia (which impacts the generalizability, the sample for the study is commendable.
I was impressed by the rigorous questionnaire development process. The researchers incorporated expert input, peer review, and a pilot study in their approach which underscores their commitment to methodological rigor. The statistical analyses employed further demonstrate the high quality of this research. The use of descriptive statistics, reliability testing, exploratory factor analysis (EFA), and confirmatory factor analysis (CFA) provides a comprehensive evaluation of the instrument's internal structure. The confirmatory factor analyses lend strong support to the adequacy of the factorial structure, while the high reliability scores indicate that the SDMH is a consistent and dependable tool.
The study's findings, which establish the SDMH as a valid and reliable measure, are not only encouraging but also pave the way for further research in this critical area. In conclusion, this study is exemplary in its design and execution.
Author Response
Thank you very much for your comments, it certainly encourages us to continue doing research with rigor and consistency. We hope that this instrument will be of great help for various scenarios in which the mental health of young people is a priority.
Reviewer 3 Report
Comments and Suggestions for Authors
Thank you for the opportunity to review this study entitled “Analysis of the Factorial Structure and Reliability of the Social Determinants of Mental Health Questionnaire for Young Adults (SDMH)” (ejihpe-3385964).
The paper explored the psychometric properties of the Social Determinants of Mental Health Questionnaire for Young Adults (SDMH). The theoretical framework was based on the Social Determinants of Health Model. Participants were 1232 young Colombians aged 18 to 28 years (M = 20.88; SD = 3.52; 46.8% were women).
In my opinion, the research topic is relevant, and the study is interesting. The sample size and the use of a solid theoretical basis are great strengths of the paper. In parallel, some issues need to be addressed before the paper will be suitable for publication.
· Abstract: the “background” section could be expanded. Currently, it is limited to only defining the objective. It could be useful to add a sentence that offers a view of the context or justifies the need for this measure.
· Introduction: to cover the background, the authors deal with the psychological consequences of COVID-19. In this regard, it would be beneficial to incorporate trend or longitudinal studies, if available. I suggest integrating some papers to present a comprehensive framework in the introduction. These could be supplemented by further literature exploration by the authors:
- Hyland et al., 2021; doi: https://doi.org/10.1016/j.psychres.2021.113905
- Gori & Topino, 2021; doi: https://doi.org/10.3390/ijerph18115651
· Method: the authors talk about “convenience sampling”, but do not provide sufficiently accurate information on the recruitment and administration procedures.
· I suggest including a CFA figure.
· I have no doubts about the results section, which seems well-structured to me.
· The authors write about “important implications for clinical practice and mental health research”, but these are only hinted at. It is important to further expand this section.
Best wishes
Author Response

(The authors gave the same response as above.)
